# V–, U–, L– or W–shaped economic recovery after Covid-19: Insights from an Agent Based Model

Dhruv Sharma[1,2]*, Jean-Philippe Bouchaud[2,3], Stanislao Gualdi[3], Marco Tarzia[4,5], Francesco Zamponi[1]

**1** Laboratoire de Physique de l'Ecole Normale Supérieure, ENS, Université PSL, CNRS, Sorbonne Université, Université de Paris, Paris, France, **2** Chair of Econophysics & Complex Systems, Ecole polytechnique, Palaiseau, France, **3** Capital Fund Management, Paris, France, **4** LPTMC, CNRS-UMR 7600, Sorbonne Université, Paris, France, **5** Institut Universitaire de France, Paris, France

* dhruv.sharma@polytechnique.org

**Data Availability Statement:** All data presented in this work can be generated using the code provided on GitLab (https://gitlab.com/sharma.dhruv/markovid).

## Abstract

We discuss the impact of a Covid-19–like shock on a simple model economy, described by the previously developed Mark-0 Agent-Based Model. We consider a mixed supply and demand shock, and show that depending on the shock parameters (amplitude and duration), our model economy can display V-shaped, U-shaped or W-shaped recoveries, and even an L-shaped output curve with permanent output loss. This is due to the economy getting trapped in a self-sustained "bad" state. We then discuss two policies that attempt to moderate the impact of the shock: giving easy credit to firms, and the so-called helicopter money, i.e. injecting new money into the households savings. We find that both policies are effective if strong enough. We highlight the potential danger of terminating these policies too early, although inflation is substantially increased by lax access to credit. Finally, we consider the impact of a second lockdown. While we only discuss a limited number of scenarios, our model is flexible and versatile enough to accommodate a wide variety of situations, thus serving as a useful exploratory tool for a qualitative, scenario-based understanding of post-Covid recovery. The corresponding code is available on-line.

## Introduction

The Covid-19 pandemic has buffeted the world economy and induced one of the most abrupt drops in output ever recorded. What comes next? Will the economy recover quickly as lockdown measures are lifted, or will the damage inflicted by the massive waves of layoffs be more permanent? In pictorial terms, will the economic crisis be V-shaped (quick recovery), as commentators were initially hoping for, or U-shaped (prolonged drop followed by a quick recovery), or perhaps W-shaped, with a relapse due either to further lockdowns, or to premature lifting of the economic support to households and firms? The possibility of an L-shaped crisis, with a permanent loss of output, is also considered. Or else, maybe, a "swoosh", with a rapid drop followed by an excruciatingly slow recovery? [1]

**Funding:** The author(s) received no specific funding for this work.

**Competing interests:** The authors have declared that no competing interests exist.

There has been a flurry of activity to understand the consequences of the economic shock due to widespread lockdowns and subsequent loss of economic activity. While some have coupled classical economic models with SIR-like epidemic models, with the underlying assumption that the economy is somehow slaved to the dynamics of Covid [2], others have reasoned in terms of traditional economic models. There has been analytical support for both quick (V- or U-shaped) recoveries [3] and prolonged (L-shaped) crisis due to a stagnation trap (poor economic forecasts leading to lower consumption leading to lower investment) [4].

There are fears of a deep recession due to a Keynesian supply shock—a deep demand shock greater in magnitude than the supply shock that caused it [5]. With many countries facing a second wave of infections, a second string of lockdowns has become necessary (an exogenous W-shaped scenario). As suggested in [6], the economic fallout from a second wave could however be mitigated by the adoption of adequate measures. Beyond the purely economic fallout of the current crisis are concerns about the broader health of the socio-political fabric of world economies after the pandemic. For small economies, it has been conjectured that a deep restructuring of their economic systems could lead to social unrest [7]. The pandemic has also kindled a re-examination of the purposes of growth and economic development that places socio-cultural transformations alongside a recalibration of economic goals [8]. Such rethinking must take into account not only the possibility of future pandemics [9], but also drive current policy decisions with an eye on a future path of sustainable growth [8, 10].

Further investigation into the economic consequences of the pandemic have taken the direction of modeling the dynamics of the pandemic to propose optimal lockdown policies [11–13]. Others have examined the impact of lock-downs on GDP growth [14]. These studies have taken the point of view of balancing a trade-off between the health of the economy and the health of the citizenry. However the evolution of the pandemic remains uncertain before vaccines become widely available, and computations based on short-term trade-offs may cause harm since policy-makers might err on the side of protecting the economy and lifting the lock-downs too early [15]. The severity of the crisis as well as the shape of the recovery will no doubt depend on many country-specific factors. Still, a qualitative understanding of the mechanisms at play and of the impact of different policies would be of great help for policy makers.

Since the seventies, a prominent school of thought in macroeconomic modeling has emphasized rationality and analytical tractability to the detriment of non-linearity and heterogeneity. Most textbook models, which are at the core of the models routinely used by Central Banks, are simple enough to be analytically tractable, yet display a rather poor phenomenology and are not fit to extreme situations. Indeed, these models are only able to describe small fluctuations around an equilibrium state of the economy, driven by mild exogenous shocks. As a consequence, they miss strongly non-linear feedback loops that may amplify mild shocks, turning them into potentially catastrophic events.

In this work, we follow a complementary path and adapt a simplified Agent Based Model (ABM), which we have studied in depth in the context of monetary policy and inflation targeting [16–18], to explore how the economic system by itself can—or not—recover from a (pandemic-induced) rapid drop of both supply and demand followed by a quick return to normal. The model is not tractable analytically, and it has to be simulated on a computer, but the simulations are extremely fast and the model can account for several important non-linear feedback channels in the economy. Note that ABMs have also been used to model the spread of the epidemic itself, see [19].

We account for the effect of the pandemic through simultaneous supply and demand shocks of various amplitudes and duration, that can be chosen to describe country specific situations. We find that U- or L-shaped recoveries occur when the economy falls into what we called a "bad phase" in [18], characterized by a self-consistently sustained state of economic

depression and deflation. When the shock is over, the time needed for the "good phase" of the economy to re-establish itself can be extremely long (so long that it might exceed the simulation time). As a function of the parameters describing the crisis (amplitude and duration of the shock), we find that there is a *discontinuous* transition between V-shape recoveries and L-shape recessions.

The length and severity of the crisis (its "typographical shape") are also strongly affected by policy measures. We thus argue that, as was done in most European states and Japan, generous policies that prevent (as much as possible) bankruptcies and redundancies should allow the economy to recover rapidly, although endogenous relapses are possible, i.e. a W-shape without a second lockdown period.

The main message of our numerical experiments is that policy should "do whatever it takes" [20] to prevent the economy tipping into such a "bad phase", taking all measures that can help the economy recover and shorten the recession period, such as "helicopter money" and easy access to credit for firms. We find that both policies are effective if strong enough, and we highlight the potential danger of terminating these policies too early. We also discuss their impact on inflation: lax access to credit helps the recovery at the expense of a higher level of inflation for an extended period of time.

Although our model is not realistic on several counts and should no doubt be enriched, we believe that it offers generic scenarios for recovery that help sharpen one's intuition and anticipate consequences that are often outside the purview of traditional approaches. Because of heterogeneities and non-linearities, collective phenomena are hard to foresee, and qualitative numerical simulations can provide a lot of insight, acting as *telescopes for the mind* [21]. Whereas ABMs are sometimes spurned because they are hard to calibrate, we have argued [16] that one should be wary of simple linear models and opt for a complementary scenario-driven approach to macroeconomic phenomena, rather than on precise-looking but possibly misleading numbers, see [22]. As Keynes famously said: *It is better to be roughly right than precisely wrong.* This is all the more so for policy makers facing a major crisis, such as the Covid-19 shock. Hence, our aim here is to provide a *qualitative* understanding of the possible recovery scenarios, but refrain from making quantitative predictions for specific situations. It is actually rewarding that several of our conclusions match those of more sophisticated Agent-Based Models that investigate the Covid-induced shock on the economy, see [23, 24].

## Methods

### Numerical simulations

Results presented here are obtained through numerically simulating the Mark-0 model [16–18]. The details of the complete model and the precise values of the parameters used are presented next.

### Description of Mark-0

The Mark-0 model with a Central Bank (CB) and interest rates has been described in full details in [16–18], where pseudo-codes are also provided. It was originally devised as a simplification of the Mark family of ABMs, developed in [25, 26]. Since the details of the model are spread across multiple papers, for purposes of completeness we summarize the fundamental aspects of the model below.

First, we establish some basic notation. The model is defined in discrete time, where the unit time between $t$ and $t + 1$ will be chosen to be $\sim 1$ month. Each firm $i$ at time $t$ produces a quantity $Y_i(t)$ of perishable goods that it attempts to sell at price $p_i(t)$, and pays a wage $W_i(t)$ to its employees. The demand $D_i(t)$ for good $i$ depends on the global consumption budget of

households $C_B(t)$, itself determined as an inflation rate-dependent fraction of the household savings. To update their production, price and wage policy, firms use reasonable "rules of thumb" [16] that also depend on the inflation rate through their level of debt (see below). For example, production is decreased and employees are made redundant whenever $Y_i > D_i$, and vice-versa. As a consequence of these adaptive adjustments, the economy is on average always "close" to the global market clearing condition one would posit in a fully representative agent framework. However, small fluctuations persists in the limit of large system sizes giving rise to a rich phenomenology [16], including business cycles. The model is fully "stock-flow consistent" (i.e. all the stocks and flows within the toy economy are properly accounted for). In particular, there is no uncontrolled money creation or destruction in the dynamics, except when explicitly stated. In our baseline simulation, the total amount of money in circulation is set to 0 at $t = 0$. This choice is actually irrelevant in the long run, but may have important short term effects. We will actually allow some money creation below, when "helicopter money" policies will be investigated.

In Mark-0 we assume a linear production function with a constant productivity, which means that output $Y_i$ and labor $N_i$ coincide up to a multiplicative factor $\zeta$, i.e. $Y_i = \zeta N_i$. The unemployment rate $u$ is defined as

$$u(t) = 1 - \frac{\sum_i N_i(t)}{N} \ , \tag{1}$$

where $N$ is the number of agents. Note that firms cannot hire more workers than available, so that $u(t) \geq 0$ at all times—see Eq (11) below. The instantaneous inflation rate, denoted $\pi(t)$, is defined as

$$\pi(t) = \frac{\bar{p}(t) - \bar{p}(t-1)}{\bar{p}(t-1)} \ , \tag{2}$$

where $\bar{p}(t)$ is the production-weighted average price $\bar{p} = \frac{\sum_i p_i Y_i}{\sum_i Y_i}$. We define $\bar{w}$ as the production-weighted wage as well: $\bar{w} = \frac{\sum_i w_i Y_i}{\sum_i Y_i}$. We further assume that households and firms adapt their behavior not to the instantaneous value of the inflation $\pi(t)$, but rather to a smoothed averaged value. We name this the "realized inflation" and it is defined as:

$$\pi^{\text{ema}} = \omega \pi(t) - (1-\omega)\pi^{\text{ema}}(t-1) \ . \tag{3}$$

Households and firms form expectations for future inflation by observing the realised inflation and the target inflation set by the Central Bank. This is modeled as follows:

$$\hat{\pi}(t) = \tau^R \pi^{\text{ema}} + \tau^T \pi^* \tag{4}$$

The parameters $\tau^R$ and $\tau^T$ model the importance of past inflation for forming future expectations and the confidence that economic actors have in the central bank's ability to achieve its target inflation.

The Mark-0 economy is made of firms producing goods, households consuming these goods, a banking sector and a central bank. Households and the banking sector are described at the aggregate level by a single "representative household" and "representative bank" respectively. In what follows, we describe how they interact within our toy economy.

**Households.** We assume that the total consumption budget of households $C_B(t)$ is given by:

$$C_B(t) = c[S(t) + W(t) + \rho^d(t)S(t)] \ , \tag{5}$$

where $S(t)$ is the savings, $W(t) = \sum_i W_i(t)N_i(t)$ the total wages, $\rho^d(t)$ is the interest rate on deposits, and $c(t)$ is the "consumption propensity" of households, which is chosen such that it increases with increasing inflation:

$$c(t) = c_0[1 + \alpha_c(\pi^{\text{ema}}(t) - \rho^d(t)] \ . \tag{6}$$

Here $\alpha_c$ is a parameter that modulates the sensitivity of households to the real interest (inflation adjusted) rate on their savings $\rho^d(t)$. With this choice of the consumption propensity, Eq (5) describes an inflation-mediated feedback on consumption similar to the standard Euler equation in DSGE models [27].

The total household savings then evolve according to:

$$S(t+1) = S(t) + W(t) + \rho^d(t)S(t) - C(t) + \Delta(t), \tag{7}$$

where $\Delta(t)$ are the dividends received from firms' profits (see below). $C(t) \leq C_B(t)$ is the actual consumption of households, determined by the matching of production and demand and computed as

$$C(t) = \sum_i^{N_F} p_i \min\{Y_i, D_i\} \leq C_B(t) = \sum_i^{N_F} p_i(t)D_i(t) \ . \tag{8}$$

The demand for goods $D_i$ itself is modeled via an intensity of choice model with a parameter $\beta$ which defines the dependence of the demand on the price:

$$D_i(t) = \frac{C_B(t)}{p_i(t)} \frac{\exp(-\beta p_i(t))}{\sum_j \exp(-\beta p_j(t))} \ . \tag{9}$$

**Firms.** The model contains $N_F$ firms (we chose $N_F = N$ for simplicity [16]), each firm being characterized by its workforce $N_i$ and production $Y_i = \zeta N_i$, demand for its goods $D_i$, price $p_i$, wage $W_i$ and its cash balance $\mathcal{E}_i$ which, when negative, is the debt of the firm. We characterize the *financial fragility* of the firm through the debt-to-payroll ratio

$$\Phi_i = -\frac{\mathcal{E}_i}{W_i N_i} \ . \tag{10}$$

Negative $\Phi$'s describe healthy firms with positive cash balance, while indebted firms have a positive $\Phi$. If $\Phi_i < \Theta$, i.e. when the flux of credit needed from the bank is not too high compared to the size of the company (measured as the total payroll), the firm $i$ is allowed to continue its activity. If on the other hand $\Phi_i \geq \Theta$, the firm $i$ defaults and the corresponding default cost is absorbed by the banking sector, which adjusts the loan and deposit rates $\rho^l$ and $\rho^d$ accordingly (see below). The defaulted firm is replaced by a new one at rate $\varphi$, initialized at random. The parameter $\Theta$ controls the maximum leverage in the economy, and models the risk-control policy of the banking sector.

*Production update*. If the firm is allowed to continue its business, it adapts its price, wages and production according to reasonable (but of course debatable) "rules of thumb" [16, 18]. In particular, the production update is chosen as follows:

$$\text{If } Y_i(t) < D_i(t) \quad \Rightarrow \quad Y_i(t+1) = Y_i(t) + \min\{\eta_i^+(D_i(t) - Y_i(t)), \zeta u_i^\star(t)\}$$

$$\text{If } Y_i(t) > D_i(t) \quad \Rightarrow \quad Y_i(t+1) = Y_i(t) - \eta_i^-[Y_i(t) - D_i(t)] \tag{11}$$

where $u_i^\star(t)$ is the maximum number of unemployed workers available to the firm $i$ at time $t$,

which depends on the wage the firm pays,

$$u^* = \frac{\exp^{\beta W_i(t)/\bar{w}(t)}}{\sum_i \exp^{\beta W_i(t)/\bar{w}(t)}} \quad, \tag{12}$$

where $\bar{w}$ is the production-weighted average wage. Firms hire and fire workers according to their level of financial fragility $\Phi_i$: firms that are close to bankruptcy are arguably faster to fire and slower to hire, and vice-versa for healthy firms. The coefficients $\eta^{\pm} \in [0, 1]$ express the sensitivity of the firm's target production to excess demand/supply. We posit that the coefficients $\eta_i^{\pm}$ for firm $i$ (belonging to $[0, 1]$) are given by:

$$\eta_i^- = [\![ \eta_0^- (1 + \Gamma \Phi_i(t)) ]\!], \tag{13}$$

$$\eta_i^+ = [\![ \eta_0^+ (1 - \Gamma \Phi_i(t)) ]\!], \tag{14}$$

where $\eta_0^{\pm}$ are fixed coefficients, identical for all firms, and $[\![ x ]\!] = 1$ when $x \geq 1$ and $[\![ x ]\!] = 0$ when $x \leq 0$. The factor $\Gamma > 0$ measures how the financial fragility of firms influences their hiring/firing policy, since a larger value of $\Phi_i$ then leads to a faster downward adjustment of the workforce when the firm is over-producing, and a slower (more cautious) upward adjustment when the firm is under-producing. $\Gamma$ itself depends on the inflation-adjusted interest rate and takes the following form:

$$\Gamma = \max[\alpha_{\Gamma}(\rho^l(t) - \pi^{\text{ema}}(t), \Gamma_0], \tag{15}$$

where $\alpha_{\Gamma}$ is a free parameter, similar to $\alpha_c$ that captures the influence of the real interest rate.

*Price update.* Following the initial specification of the Mark series of models [25], prices are updated through a random multiplicative process, which takes into account the production-demand gap experienced in the previous time step and if the price offered is competitive (with respect to the average price). The update rule for prices reads:

$$
\begin{aligned}
\text{If } Y_i(t) < D_i(t) \quad \Rightarrow \quad &
\begin{cases}
\text{If } p_i(t) < \bar{p}(t) & \Rightarrow \quad p_i(t+1) = p_i(t)(1 + \gamma \xi_i(t))(1 + \hat{\pi}(t)) \\
\\
\text{If } p_i(t) \geq \bar{p}(t) & \Rightarrow \quad p_i(t+1) = p_i(t)
\end{cases} \\
\\
\text{If } Y_i(t) > D_i(t) \quad \Rightarrow \quad &
\begin{cases}
\text{If } p_i(t) > \bar{p}(t) & \Rightarrow \quad p_i(t+1) = p_i(t)(1 - \gamma \xi_i(t))(1 + \hat{\pi}(t)) \\
\\
\text{If } p_i(t) \leq \bar{p}(t) & \Rightarrow \quad p_i(t+1) = p_i(t)
\end{cases}
\end{aligned}
\tag{16}
$$

where $\xi_i(t)$ are independent uniform $U[0, 1]$ random variables and $\gamma$ is a parameter setting the relative magnitude of the price adjustment. The factor $(1 + \hat{\pi}(t))$ models firms' anticipated inflation $\hat{\pi}(t)$ when they set their prices and wages.

*Wage update.* The wage update rule follows the choices made for price and production. Similarly to workforce adjustments, we posit that at each time step firm $i$ updates the wage paid to its employees as:

$$
\begin{aligned}
W_i^T(t+1) = W_i(t)[1 + \gamma(1 - \Gamma \Phi_i)(1 - u(t))\xi_i'(t)][1 + g\hat{\pi}(t)] \quad \text{if } &
\begin{cases}
Y_i(t) < D_i(t) \\
\\
\mathcal{P}_i(t) > 0
\end{cases} \\
\\
W_i(t+1) = W_i(t)[1 - \gamma(1 + \Gamma \Phi_i)u(t)\xi_i'(t)][1 + g\hat{\pi}(t)] \quad \text{if } &
\begin{cases}
Y_i(t) > D_i(t) \\
\\
\mathcal{P}_i(t) < 0
\end{cases}
\end{aligned}
\tag{17}
$$

where $\mathcal{P}_i$ is the profit of the firm at time $t$, $\xi'_i(t)$ an independent $U[0, 1]$ random variable and $g$ modulates how wages are indexed to the firms' inflation expectations. If $W_i^T(t + 1)$ is such that the profit of firm $i$ at time $t$ with this amount of wages would have been negative, $W_i(t + 1)$ is chosen to be exactly at the equilibrium point where $\mathcal{P}_i(t) = 0$; otherwise $W_i(t + 1) = W_i^T(t + 1)$. Here, $\Gamma$ is the same parameter introduced in Eq (13).

Note that within the current model the productivity of workers is not related to their wages. The only channel through which wages impact production is that the quantity $u_i^\star(t)$ that appears in Eq (11), which represents the share of unemployed workers accessible to firm $i$, is an increasing function of $W_i$. Hence, firms that want to produce more (hence hire more) do so by increasing $W_i$, as to attract more applicants.

The above rules are meant to capture the fact that deeply indebted firms seek to reduce wages more aggressively, whereas flourishing firms tend to increase wages more rapidly:

- If a firm makes a profit and it has a large demand for its good, it will increase the pay of its workers. The pay rise is expected to be large if the firm is financially healthy and/or if unemployment is low because pressure on salaries is high.

- Conversely, if the firm makes a loss and has a low demand for its good, it will attempt to reduce the wages. This reduction is more drastic if the company is close to bankruptcy, and/or if unemployment is high, because pressure on salaries is then low.

- In all other cases, wages are not updated.

*Profits and dividends*. Finally, the profits of the firm $\mathcal{P}_i$ are computed as the sales minus the wages paid with the addition of the interest earned on their deposits and the interest paid on their loans:

$$\mathcal{P}_i = p_i(t) \min\{Y_i(t), D_i(t)\} - W_i(t)Y_i(t) + \rho^d \max\{\mathcal{E}_i(t), 0\} - \rho^l \min\{\mathcal{E}_i(t), 0\} \ . \quad (18)$$

If the firm's profits are positive and they possess a positive cash balance, they pay dividends as a fraction $\delta$ of their cash balance $\mathcal{E}_i$:

$$\Delta(t) = \delta \sum_i \mathcal{E}_i(t)\theta(\mathcal{P}_i(t))\theta(\mathcal{E}_i(t)) \ , \quad (19)$$

where $\theta$ is the Heaviside step-function. These dividends are then reduced from the firms' cash balance and added to the households savings in Eq (7).

**Banking sector.** The banking sector in Mark-0 consists of one "representative bank" and a central bank which sets baseline interest rates. The central bank also has an inflation targeting mandate. The central bank sets the base interest rate $\rho_0$ via a Taylor-like rule:

$$\rho_0(t) = \rho^* + \phi_\pi[\pi^{\text{ema}} - \pi^*] \ . \quad (20)$$

Here, $\phi_\pi$ modulates the intensity of the central bank policy, $\rho^*$ is the baseline interest rate and $\pi^*$ is the inflation target for the central bank. The banking sector then sets interest rates for deposits $\rho^d$ (for households) and loans $\rho^l$ (for borrowing by firms). Defining $\mathcal{E}^+ = \sum_i \max(\mathcal{E}_i, 0)$ (equivalent to firms' positive cash balance) and $\mathcal{E}^- = -\sum_i \min(\mathcal{E}_i, 0)$ (equivalent to firms' total debt), the interest rates are set as:

$$\rho^l(t) = \rho_0(t) + f\frac{\mathcal{D}(t)}{\mathcal{E}^-(t)}, \quad (21)$$

$$\rho^d(t) = \frac{\rho_0 \mathcal{E}^-(t) - (1-f)\mathcal{D}(t)}{S + \mathcal{E}^+(t)},$$ (22)

where $\mathcal{D}(t)$ are the total costs accrued due to defaulting firms. The parameter $f$ then determines how the impact of these defaults fall upon lenders and depositors—$f$ interpolates between these costs being borne completely by the borrowers ($f = 1$) or fully by the depositors ($f = 0$). The total amount of (central-bank) money $M$ in circulation is kept constant and the balance sheet of the banking sector reads:

$$M = S(t) + \mathcal{E}^+(t) - \mathcal{E}^-(t) \ .$$ (23)

This is simply a restatement of the fact that the sum of households savings and the deposits and debts of firms is equal to the total amount of money in circulation.

## Summary

The model, as presented above, has a rich phase diagram and even though it possesses many free parameters, it has been found that only $R$ (hiring/firing ratio) and $\Theta$ (bankruptcy threshold for firms) are important [16, 18]. The model hence presents four phases in the $\Theta - R$ plane:

- Full Employment (FE) phase: characterized by positive average inflation and close to full employment,

- Full Unemployment (FU) phase: characterized by high unemployment and deflation,

- Residual Unemployment (RU) phase: characterized by zero inflation but with some residual unemployment, and

- Endogenous crises (EC) phase: characterized by low unemployment and inflation on average, but with the intermittent spikes of "endogenous crises" accompanied by high unemployment and deflation.

The set of parameters fixed to establish the baseline scenario are presented in Table 1. With these parameters, the economy is in the Full Employment (FE) phase.

In our examination of the Covid-induced shocks to our toy economy, certain simplifying assumptions have been made:

**Table 1. Parameters of the Mark-0 model relevant for this chapter, together with their symbol and baseline values.**

| | | |
|---|---|---|
| Number of firms | $N_F$ | 10000 |
| Consumption propensity | $c_0$ | 0.5 |
| Intensity of choice parameter | $\beta$ | 2 |
| Price adjustment parameter | $\gamma$ | 0.01 |
| Firing propensity | $\eta_-^0$ | 0.2 |
| Hiring propensity | $\eta_+^0$ | $R\eta_-^0$ |
| Hiring/firing ratio | $R$ | 2 |
| Fraction of dividends | $\delta$ | 0.02 |
| Bankruptcy threshold | $\Theta$ | 3 |
| Rate of firm revival | $\varphi$ | 0.1 |
| Productivity factor | $\zeta$ | 1 |
| Financial fragility sensitivity | $\Gamma_0$ | 0 |
| Exponentially moving average (ema) parameter | $\omega$ | 0.2 |

1. The inflation expectations are set to zero, i.e. in Eq (4), $\tau^R$ and $\tau^T$ are both set to zero.

2. We neglect the feedback of inflation on consumption i.e. $\alpha_c = 0$ in Eq (6). This implies that the consumption propensity $c$ is constant in time, and equal to $c_0$.

3. Firms' financial fragility doesn't affect their hiring/firing rates i.e. $\alpha_\Gamma$ and $\Gamma_0$ are set to zero in Eq (15).

We also make the strong assumption of not using the traditional monetary policy channel, i.e. the central bank is switched off—the base interest rate $\rho_0 = 0$ and the central bank performs no inflation targeting, $\pi^* = 0$.

Finally, we stress that because the dynamics of the Mark-0 model depends, in several instances, on random numbers—e.g. in the rule for production and wage update—the results of single simulations runs are, *a priori*, sensitive to noise. We repeated each simulation multiple times with different seeds of the random number generator and we found that all our results are qualitatively robust, and only the details are sensitive to the noise. As an example of this sensitivity, we note that for the scenarios with a W-shaped recovery reported in Fig 5, the presence of the second downturn is not observed in all runs.

In what follows, we fix a certain number of parameters to establish a baseline scenario, in the absence of shocks. With these parameter values, the unemployment level is low and the economy is at near maximal output (with productivity $\zeta = 1$). The annual inflation rate is $\approx$ 1.3%/year (similar to the annual inflation rate in the Eurozone) and the average financial fragility $\langle \Phi \rangle$ of the firms is $\approx 1$ (i.e. debt equal to one month of wages), far from the baseline bankruptcy threshold $\Phi \leq \Theta = 3$. Finally, households consumption propensity, which models the fraction of their total savings they consume at each step, is set to $c = 0.5$. Other parameters have been summarized in Table 1.

Let us insist that these parameter values are somewhat arbitrary but can be changed at will to reproduce more faithfully the state of the economy of a particular country before the Covid crisis. The results given below are meant to illustrate some generic properties of our model which, quite interestingly, match qualitatively with those obtained in more sophisticated ABMs [24].

## Results

The impact of the pandemic on the economy has been through a fall in consumption—since countries are under lockdown or there is widespread fear of the disease and people stay home—and a simultaneous loss in productivity—since people stay home and firms are unable to maintain production [28]. To understand such Covid-19–induced dual consumption and productivity shocks, we study the Mark-0 model described in all details in [16–18] and in the Methods section above. This stylized Agent-Based Model (ABM) is simple enough that a trajectory of the economy over a time span of a few decades can be simulated in a few seconds on a standard workstation. Many parameters can be changed, such as the length and severity of the shock, the amplitude of the policy response, the confidence of households, etc. In the present paper, we have only explored a small swath of possibilities for a limited subset of the possible feedback channels in the model. In order to allow our readers to reproduce our results, and explore the variety of possible outcomes via additional experiments, our code is freely accessible on GitLab. We discuss possible extensions of our results in the Conclusions section below.

### Simulating shocks to the economy

As mentioned above, a lockdown leads to a drop in both supply and demand, corresponding in our model to a sudden drop in both the productivity of firms, $\zeta \rightarrow \zeta - \Delta\zeta$, and in the consumption propensity of households, $c \rightarrow c - \Delta c$.

Given that the infection continues to spread, it is uncertain how long the effects of the crisis will last. Hence, an important parameter describing the shock is its effective duration $T$. In what follows, we choose three values for the duration of the first shock: 3 months, 6 months and 9 months. Lockdown measures can be partially lifted which may lead to an increased value for both $\zeta$ and $c$ during the shock period. We also note that although lockdowns might not last for as long as 9 months, the effects of the pandemic on consumer behavior might persist for longer. As has already been noted, the economic impact on people's lives may last well beyond the lifting of the lockdowns with consumer spending lower than pre-pandemic levels [29]. A drop of consumption propensity could reflect a long lasting loss of confidence, for example. The values for $T$ are thus meant to represent an effective length of the pandemic-induced shock. We also consider explicitly a scenario with a second lockdown associated to a second wave of the epidemics.

## Phenomenology

We begin with an exploration of the various possible scenarios for the evolution of the economy following the shock. In Fig 1, we show several typical crisis and recovery shapes, depending on the strength of the shock and the policy used to alleviate the severity of the crisis. For small enough $\Delta c$ and/or $\Delta \zeta$, one observes a V-shape recovery, as expected when the shock is mild enough not to dent the financial health of the firms. Stronger shocks can however lead to a permanent dysfunctional state (L-shape), with high unemployment, falling wages and savings, and a high level of financial fragility and bankruptcies. An L-shaped scenario can however be prevented if after the shock, consumer demand rises i.e by increasing $c$. To facilitate and boost consumption, a one-time policy of helicopter money can move the economy towards a path of recovery over the scale of a few years (U-shape). Interestingly, for shocks lasting for nine months, a one-time helicopter drop of money without any other interventions can lead to a W-shaped recovery.

## Phase diagrams

To better understand the influence of these shocks, we then carry on a more systematical investigation of the model's behavior in the phase diagram defined by the control parameters

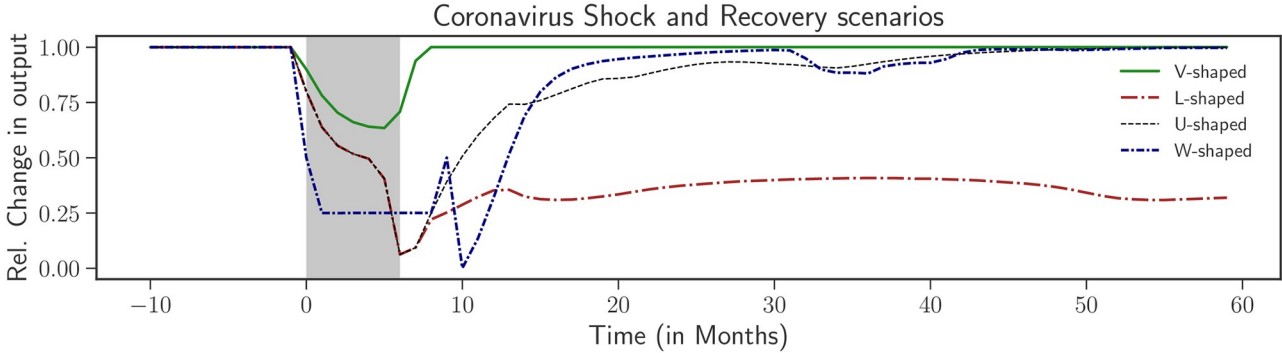

**Fig 1. Possible recovery patterns following the coronavirus shock, which starts at time $t = 0$.** We show, as a function of time, the fall in output relative to the no-shock scenario. For mild shocks ($\Delta c/c = 0.3$, $\Delta \zeta/\zeta = 0.1$, lasting six months), the economy contracts but quickly recovers (V-Shaped recovery). For more severe shocks over the same time period ($\Delta c/c = 0.3$, $\Delta \zeta/\zeta = 0.2$), the economy contracts permanently and never recovers (at least on the time scale of the simulation). This is the dreaded L-shaped scenario, in the absence of any government policy. An increase in consumption propensity to $c = 0.7$ (0.2 above pre-crisis levels) and a helicopter money drop at the end of the shock leads to a faster recovery (U-shaped recovery). Finally, a W-shaped scenario, with a second downturn occurring 30 months after the initial shock, can also be found for stronger shocks ($\Delta c/c = 0.3$, $\Delta \zeta/\zeta = 0.5$, lasting nine months) with strong government policy and a helicopter drop.

$\Delta c/c$, $\Delta\zeta/\zeta$, for $T$ = 3, 6 and 9 months in Fig 2. The phase diagrams are all constructed in the absence of policy, because we want to first understand in which regions of phase-space policy would indeed be necessary. Within each phase diagram, we present (a) the probability of a dire (L-shaped) crisis, (b) the peak value of unemployment during the shock and (c) the peak value of unemployment after the shock. Yellow/green regions indicate that the economy recovers well—no dire crisis, or short crises with low levels of unemployment. This occurs, as expected, for small shocks, small $\Delta c/c$, $\Delta\zeta/\zeta$, in the lower left corner of the graphs. These regions shrink as $T$ increases implying that even mild shocks, but sustained over a longer period of time, can lead to dire crises. We note that mild shocks lasting only for a short time ($T$ = 3 months) can also cause lasting damage. Indeed, we observe that low rates of unemployment during the shock are not representative of future evolution. Interestingly, there is an abrupt, first order transition line (a "tipping point" in the language of [16]) beyond which crises have a very large probability to become permanent, with high levels of unemployment. In the real economy, the precise location of such tipping points is hard to estimate. We are led to conclude that governments and other institutions should err on the side of caution with their policy making. Note that, interestingly, similar tipping points appear in the ABM studied in [24].

We can also specialize on the $\Delta\zeta = 0$ line of these two-dimensional plots, corresponding to a consumption shock without a simultaneous productivity shock. We show in Fig 3 the same three quantities as in Fig 2. An abrupt transition between no dire crises and dire crises can be seen for $\Delta c/c \sim 0.4$ when $T$ = 9 months.

## Policy proposals for quick recovery

The toolbox developed in the aftermath of the Global Financial Crisis (GFC) of 2008 puts monetary policy at the center of economic crisis management (see [17, 18] for a discussion of monetary policy within the Mark-0 model). This takes the form of either direct interest rate cuts or, as was seen recently for interventions following the GFC, even stronger measures such as quantitative easing. Given that the interest rates are already very low, the interest-rate channel itself might not be effective, and might lead to a stagnation trap and a L-shaped recovery [4]. Hence, because monetary policy cannot be used as an emergency measure in the face of a collapsing supply sector, for simplicity we disregard in this paper the interest-rate channel as a possible policy tool. Of course, for the longer term fate of the economy, the monetary policy is without any doubt important, and it can easily be switched back on in our model, following [18]. We leave this extension for future work.

Consequently, we discuss two other policy channels: easy credit for firms, and "helicopter money" for households.

**Policy-1: Easy credit access to firms.** A way to loosen the stranglehold on struggling firms is to give them easy access to credit lines, independently of their financial situation. Within our model, this is equivalent to an increase of the bankruptcy threshold $\Theta$. The policy we investigate is then the following: during the whole duration of the shock, we set $\Theta = \infty$—all firms are allowed to continue their business as usual and accumulate debt. When the shock is over, the value of $\Theta$ is reduced to its pre-crisis levels i.e. $\Theta = 3$. Tuning down the credit access line can be done in multiple ways. One extreme possibility, termed "naive" below, is to set $\Theta$ to its pre-shock value as soon as the shock is over. Intuitively, when the shock is short enough, allowing endangered firms to survive might be enough. For longer shocks, such a naive policy might not be helpful because firms that have pulled through the crisis have become more fragile (accumulated too much debt) by the end of the shock. In this scenario, many firms fail when credit is tightened, and the economy plunges into recession as if no policy were applied.

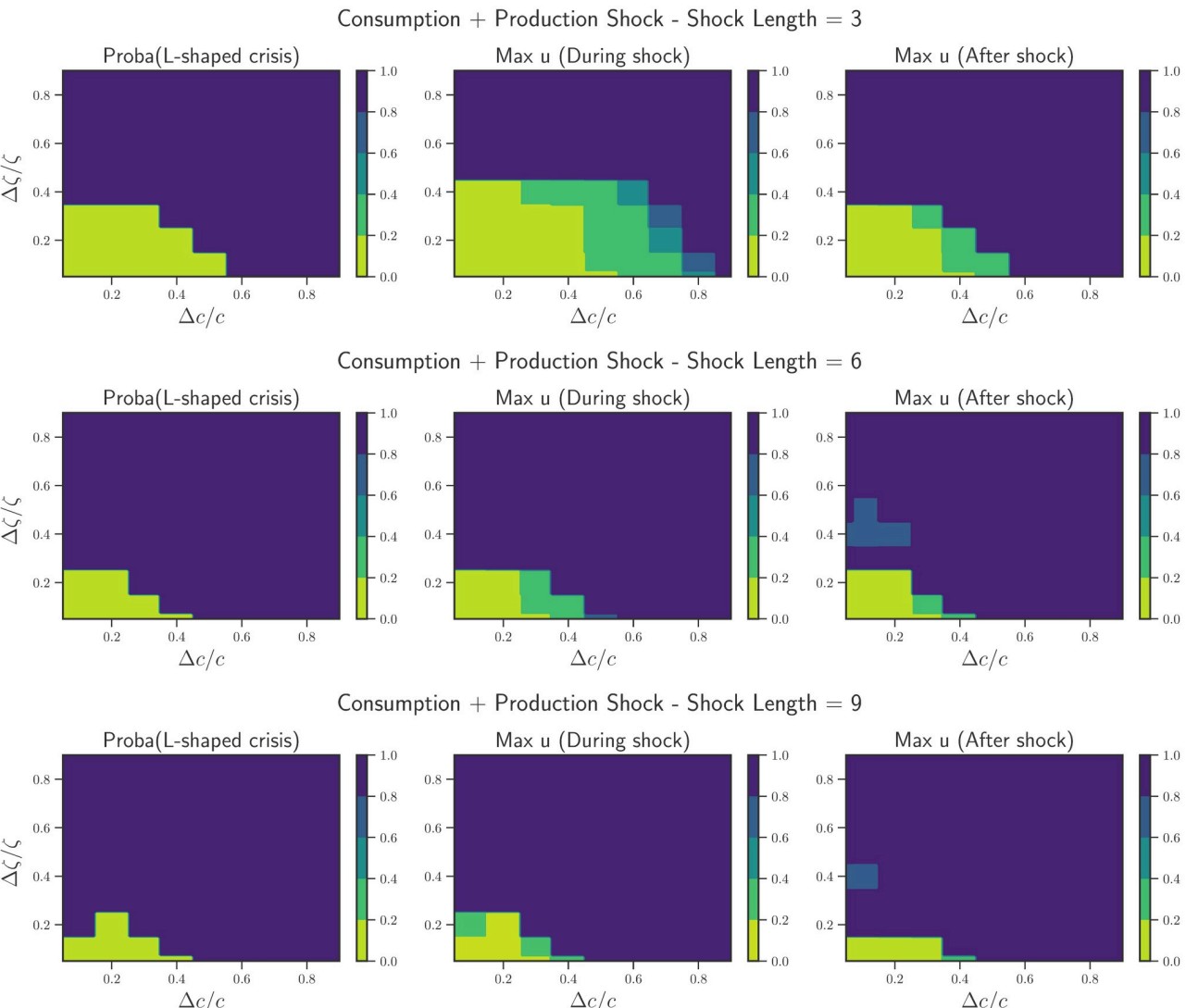

**Fig 2. Phase diagrams in the Δc/c—Δζ/ζ plane for different shock lengths.** *Top Row*: Shock lasting for 3 months. The region of parameter space with no L-shaped crisis is quite large allowing for strong consumption shocks ($\Delta c/c \lesssim 0.5$) and mild productivity shocks ($\Delta\zeta/\zeta \lesssim 0.3$). Note that for such a short shock, the effects on unemployment are seen after the shock has ended. *Middle Row*: Shock lasts for 6 months. A decrease in the region of no-crisis is observed. Mild shocks ($\Delta c/c \sim 0.4$) can also lead to prolonged crises. During the shock itself, extremely high rates of unemployment can be seen. *Bottom Row*: Shock lasts for 9 months. Only for extremely mild shocks does the economy not undergo a prolonged crisis. Each of the plots has been obtained by running 500 independent simulations for each pair $\Delta\zeta/\zeta$—$\Delta c/c$ presented here.

Another possibility, that we call "adaptive", is to reduce Θ progressively, in a way that is adapted to firms' average fragility: the government measures the instantaneous value of ⟨Φ⟩ averaged over firms still in activity weighted by production $Y_i$, i.e. $\langle\Phi\rangle = \Sigma_i \Phi_i Y_i / \Sigma_i Y_i$, and sets Θ as:

$$\Theta = \max(\theta\langle\Phi\rangle, 3), \qquad (t > T), \tag{24}$$

where $\theta$ is some offset that we chose to be $\theta = 1.25$. This means that only the most indebted firms, whose fragility exceeds the average value by more than 25%, will go bankrupt as the effective threshold Θ is progressively reduced.

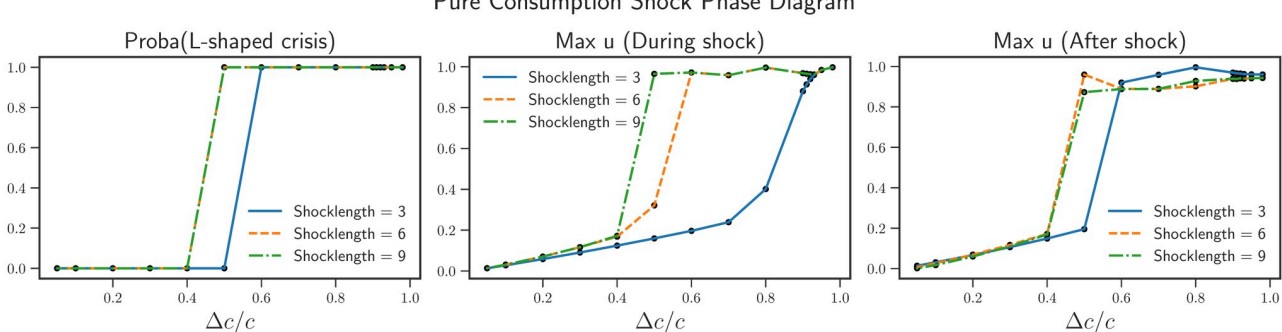

**Fig 3. Phase diagram for a pure consumption shock ($\Delta\zeta/\zeta = 0$).** We observe an abrupt transition to an L-shaped crisis for consumption shocks beyond $\Delta c/c = 0.5$ for $T = 3$ months, and beyond $\Delta c/c = 0.4$ for $T = 6$ or 9 months. Below these shock amplitudes, there is no prolonged crises but short-lived crises are observed. This can be seen by observing the maximum unemployment rate during the shocks: for $\Delta c/c = 0.4$, we reach about 15% unemployment. Each of the plots has been obtained by running 500 independent simulations for each value of $\Delta c/c$ presented here.

A primary reason for focusing on firms' credit limits and not the banking sector directly is that the major risks from the Covid-19 shock are not going to be transmitted via a systemic crisis in the financial sector. It is the private non-banking sector which is facing the brunt of the health measures such as lockdowns. As argued in [30], 2020 is not 2008—while 2008 was an endogenous crisis transmitted through the interlinked nature of modern finance, the Covid-19 shock is well and truly exogenous. Furthermore, given the experience of the 2008 crisis, central banks and other monetary institutions are better equipped to prevent any systemic risks within the financial sector—they have indeed "flattened the curve" of financial panic [31]. Finally, a comparison with the 2008 crisis shows that a key factor determining future economic growth are retail sales [32]. Hence we focus on a policy which allows firms' to stay afloat and weather the economic shock.

**Policy-2: "Helicopter money".** Another possibility for mitigating losses due to the pandemic is for the governments to take a more active role. Governments can undertake financing for emergency requirements by issuing debt. However, this mechanism leaves future governments vulnerable to interest rate changes [33]. An alternative is to inject cash in the economy to boost consumption and facilitate recovery [34]. This involves the expansion of the money supply by the central bank with the newly created money lent directly to the government. The central bank then immediately writes off this "loan". The government can then spend it on emergency healthcare requirement or other infrastructure projects. The distribution of the newly-created reserves has to be intermediated via the banking sector.

To overcome this, following Friedman's original proposal [35], central banks can provide the money directly to households. This is the version of a "helicopter money" drop examined here. This policy has been considered radical due to the fear that an expansion in money supply might lead to runaway inflation. In normal times, there might be support for such a view, but it has been shown that a helicopter drop may not always be inflationary [36]. Given the enormity of the crisis, there have been calls from all corners for central banks to break "taboos" [20, 34, 37, 38] and do what is necessary.

In this work, we implement a helicopter-money drop by assuming that the government distributes money to households multiplying their savings by a certain factor $\kappa > 1$: $S \to \kappa S$. The distribution takes place at the end of the shock, and we study here how the "naive policy" (for which $\Theta$ goes back to its baseline value immediately after the shock) can be improved by some helicopter money. In what follows $\kappa = 1.5$, i.e. households' savings are increased by 50% via a one-time money injection.

Of course, the efficacy of helicopter money relies on the consumption propensity $c$ of households. Here, we assume that $c$ recovers its pre-shock value immediately, but a loss of confidence may lead to persistently low values of $c$, weakening the effect of the policy. The effect of a slow recovery of confidence, through a coupling between the value of $c$ and unemployment, could easily be included in the model, following [17, 18].

**Implementation of policies.** We now implement these governmental policies. In what follows, we choose $\Delta c/c = 0.3$ and $\Delta\zeta/\zeta = 0.5$ as reasonable values to represent the severity of the Covid-19 shock [39, 40], and $T$ takes values 3 and 9 months. In Figs 4 and 5 we present a dashboard of the state of the economy: output and unemployment, financial fragility and the bankruptcy rate, inflation and wages, savings and interest rates. In each case, we present four scenarios: a baseline scenario without any policy in the first column ("No policy"), the scenario with the naive policy in the second column ("Naive policy"), with a "helicopter drop" in the third column ("Naive policy + Helicopter") and the situation with the adaptive policy in the fourth column ("Adaptive policy"). In all cases, in the absence of any active policy, the economy collapses into a deep recession, with an output reduced by 2/3 compared to pre-shock levels. The trajectories we show correspond to a single run of the model, but different runs yield qualitatively similar results.

It is instructive to remark how the duration of the shock influences which policy is successful. For a shock that lasts for 3 months, presented in Fig 4, we observe that the economy contracts and remains in a depressed state. A significant loss in real wages is observed, accompanied by a high unemployment rate. We also observe that a large number of firms go bankrupt and households savings are reduced permanently. Finally, with bankruptcy rate quite high, the interest rate on loans $\rho^l$ also sees a dramatic increase. This L-shaped scenario can be improved by extending the credit limits of the firms for the length of the crisis (naive policy). This improves the situation of the economy even though a temporary contraction in output can not be avoided (V-shaped recovery). The other two policies—"helicopter money" and adaptive policy—perform similar to the naive policy. Note that at the end of the shock, when $c$ returns to its original value, households start to over-spend with respect to the pre-crisis level, because their savings increase during the shock (mirroring the increase of firms' debt) and they want to spend a fixed fraction of them. However, this over-spending alone can be insufficient to drive back the economy to its pre-crisis state.

The situation for a longer shock duration—9 months, shown in Fig 5—is markedly different. As before, without any policy intervention, the economy suffers a severe and prolonged contraction, similar to the case for a shock duration of 3 months. However, given the length of the shock, there is a deeper fall in the level of real wages with firms continuing to go bankrupt far after the shock has ended. The introduction of the naive policy in this case is unable to rescue the economy—extending credit limits to the firms during the crisis prevents them from going bankrupt at first, but as soon as the policy is removed, these indebted firms fail and unemployment shoots up drastically, with further downward pressure on the wages.

The introduction of helicopter money improves upon the naive policy. Nonetheless, an interesting effect appears, in the form of a W-shape, or relapse of the economy, *even in the absence of a second lockdown period*. This "echo" of the initial shock is due to financially fragile firms that eventually have to file for bankruptcy when credit lines are tightened post-shock. This second downturn is however temporary and the economy manages to recover fairly quickly. This experiment shows the importance of boosting consumption when the shock is over. A similar effect would be obtained if instead of the savings $S$, the consumption propensity $c$ was increased post-lockdown. A combination of the two might indeed lead the economy to a faster recovery as shown in the U-shape recovery in Fig 1.

Consumption + Production Shock - Shocklength = 3, $\Delta c/c = 0.3$, $\Delta \zeta/\zeta = 0.5$

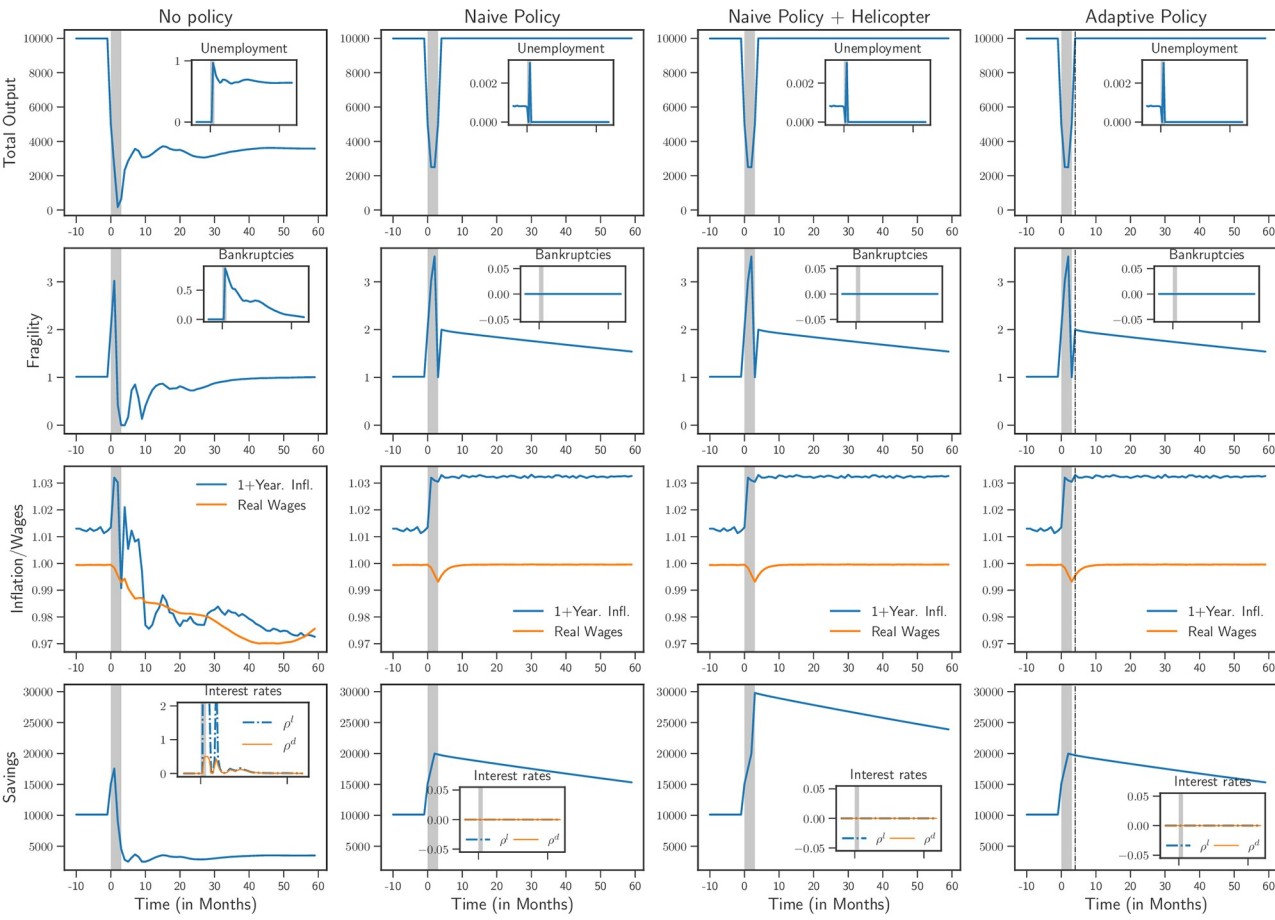

**Fig 4. Scenarios for shock length of 3 months with and without policy is shown.** *First column*: Without any policy, the economy suffers a deep contraction. *Second column*: The introduction of the naive policy improves the situation of the economy and a V-shaped recovery is observed. The results for the other two policies (*third and fourth columns*) are equivalent to those with the naive policy. Note that these successful policies leads to a markedly increased inflation. The inflation returns to the pre-crisis level only when savings recover their pre-crisis level too.

Finally, the adaptive policy is indeed the most successful one. A contraction during the crisis is inevitable but the economy recovers 100% of its pre-shock output at the end of the shock. As can be seen from the plot of the average fragility, this comes at the price of $\langle \Phi \rangle$ reaching very high values for a while—for example $\langle \Phi \rangle \approx 6$—and, much higher post-crisis inflation ($\approx$ 3%). But the slow removal of the easy credit policy allows the economy to revert smoothly to its pre-shock state, with bankruptcies kept low. It is important to note that this policy needs to be implemented for a rather long period of time after the shock (almost 7 years in our simulations).

We also studied the case of a pure, rather severe consumption shock $\Delta c/c = 0.7$ lasting $T = 9$ months, shown in Fig 6. We observe that a prolonged drop in consumption, without any loss in production, can still lead to long-lived crisis. The "naive" policy in this case is not enough to hasten the recovery. Direct cash transfer to households via helicopter money drop helps the economy recover faster but leads to a slow, W-shape recovery. Finally, the "adaptive" policy again works best in keeping unemployment low and ensures a rapid recovery, accompanied by higher inflation.

Consumption + Production Shock - Shocklength = 9, $\Delta c / c = 0.3$, $\Delta \zeta / \zeta = 0.5$

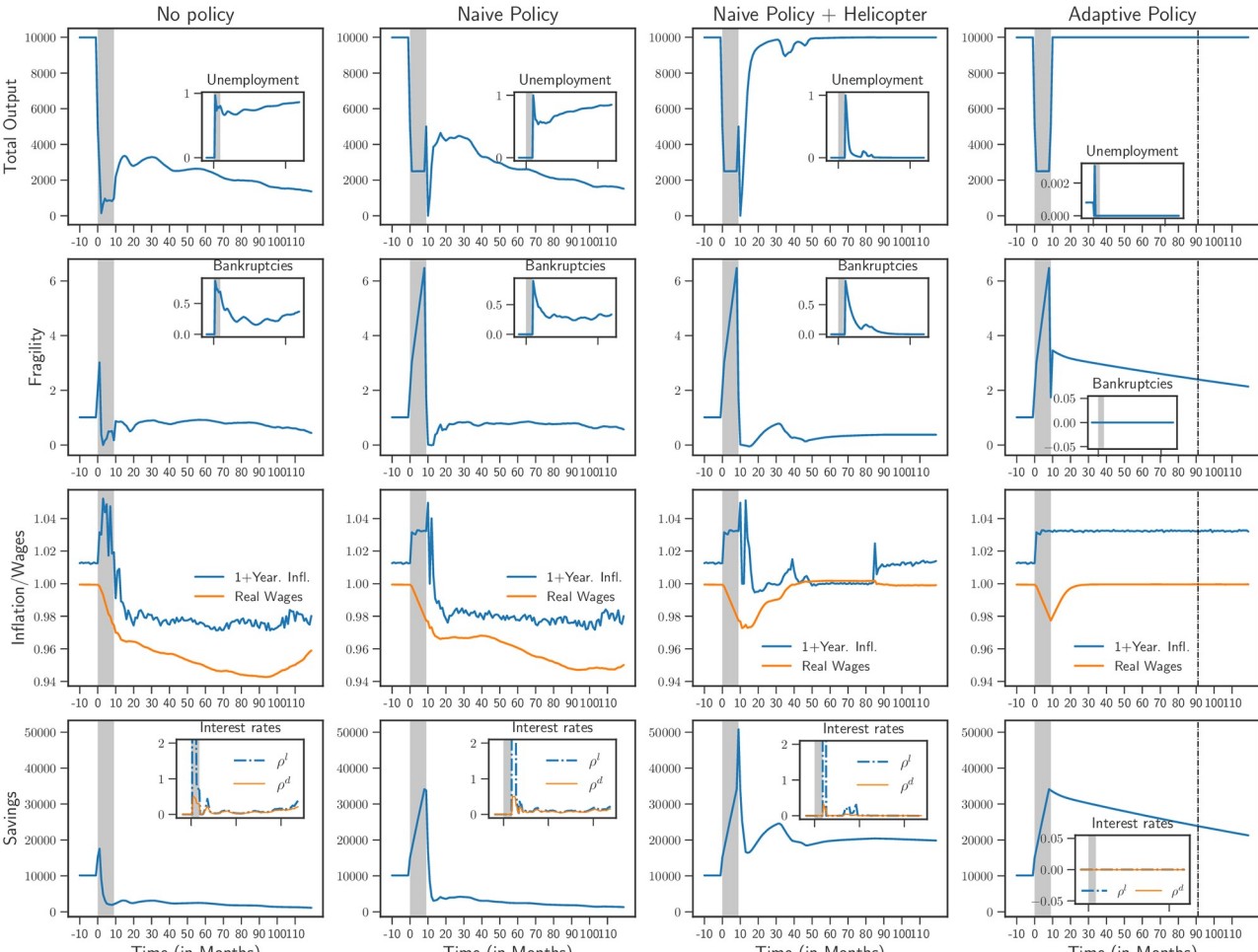

**Fig 5. Scenarios for shock length of 9 months (marked in grey) with and without policy.** *First column*: Similar to Fig 4, the economy undergoes a severe and prolonged contraction. However, given the length of the shock, there is a deeper fall in the level of real wages with firms continuing to go bankrupt far after the shock has occurred. The naive policy (*Second column*) alone is not successful but the introduction of helicopter money (*Third column*) is able to rescue the economy at the expense of a second milder "echo" shock later. Finally, the adaptive policy (*Fourth column*) is able to achieve a smooth recovery with little to no bankruptcies and low unemployment, but a markedly higher inflation, which returns back to pre-crisis levels when savings also return to equilibrium.

**Multiple lockdowns.** While we have discussed the possibility of an endogenous W-shaped scenario in Fig 5, we now discuss how multiple lockdowns affect economic outcomes. Indeed, in many countries, a second series of lockdowns has become unavoidable as the number of Covid-19 cases spiralled out of control in the fall of 2020.

In what follows, we briefly discuss a simple scenario with two lockdowns, each lasting 3 months, with the lockdown being lifted for 4 months between them. We make the further assumption that after the first lockdown, consumption patterns for households and firm productivity do not instantaneously go back to their pre-lockdown values, but recover in a linear fashion. Once again, we test the four policies discussed above and compare the outcome with the case of a single lockdown. Results are presented in Fig 7.

With two lockdowns, the situation without any policy intervention remains dire and the economy is found in a permanently depressed state. There is a small uptick in output during

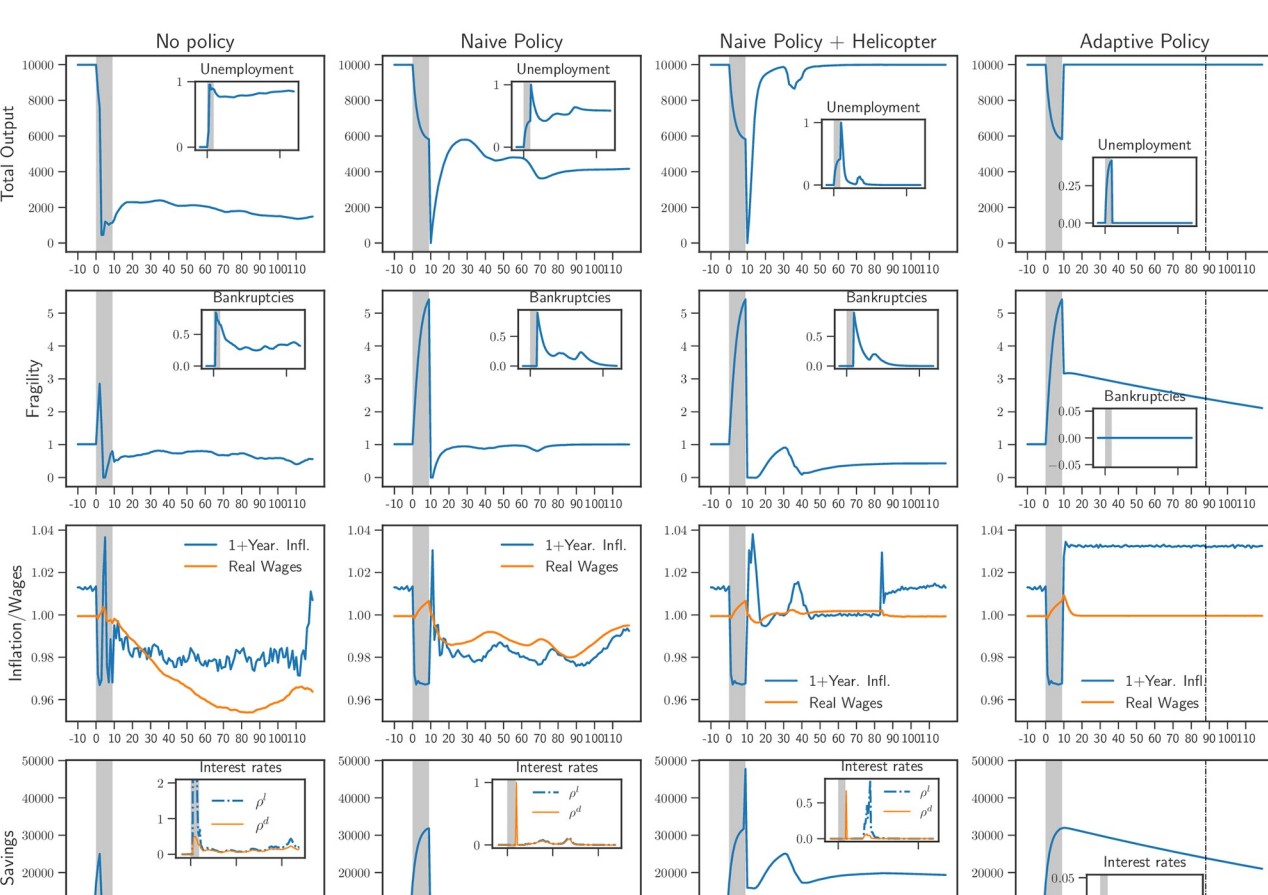

**Fig 6. Scenarios for a severe consumption shock Δc/c = 0.7 lasting for 9 months (marked in grey) with and without policy for a pure consumption shock.** Similarly to the situation shown in Fig 5, a prolonged contraction is observed, which is best alleviated by an adaptive policy.

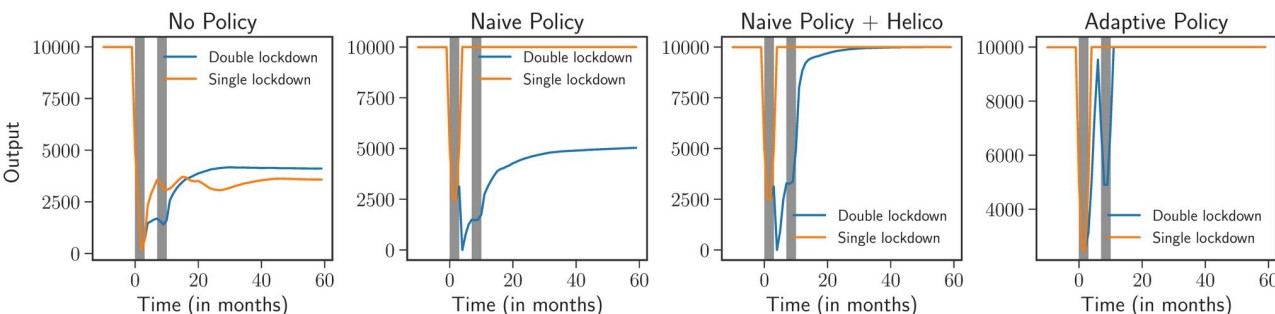

**Fig 7. Comparison between scenarios following a single lockdown of 3 months and two lockdowns of 3 months with a 4 month lockdown free period.** The naive policy, which is able to stabilize the economy in the former case, is unable to do so in the latter. Directly boosting households' savings via helicopter drops improves upon the naive policy. The adaptive policy does best and we observe an exogenous W-shaped scenario.

the lockdown free months, but at the end of the second series of lockdowns, the economy is pushed into a "L-shaped" scenario. The naive policy, when put in place for just the first 3 months, is not sufficient for a quick recovery, even though such a policy was enough for a rapid recovery in the single lockdown case (Fig 4). With consumption habits and productivity coming back only slowly, the naive policy is no longer enough for a quick recovery. Boosting households' savings via helicopter drops helps the economy recover and improves upon the naive policy.

Finally, the best policy is still the adaptive one, which extends supports to firms in guaranteeing that the economy does not tip into a permanently bad state. Interestingly, we observe a W-shaped recovery which is now due to exogenous reasons, namely a second wave of the disease.

## Discussion & conclusion

In this paper, we have proposed a qualitative assessment of the impact of Covid-19 on the economy, based on the Mark-0 Agent-Based Model. We have shown that, depending on the amplitude and duration of the shock, the model can describe different kinds of recoveries (V-, U-, W-shaped), or even the absence of full recovery (L-shape). Indeed, as discussed in [16], the non-linearities of Mark-0 allow for the presence of "tipping points" (or phase transitions in the language of physics), for which infinitesimal changes of parameters can induce macroscopic changes of the economy. The model displays a self-sustained "bad" phase of the economy, characterized by absence of savings, mass unemployment, and deflation. A large enough shock can bring the model from a flourishing economy to such a bad state, which can then persist for long times, corresponding to decades in our time units. Whether such a bad phase is truly stable forever or would eventually recover (via a "nucleation" effect similar to meta-stable phases in physics) is an interesting conceptual point. It could also have practical implications because if recovery happens via nucleation, one could imagine triggering it via the artificial creation of the proper "nucleation droplet", in the physics parlance. We leave this discussion for future studies.

We have also examined how government policies can prevent an economic collapse. For simplicity, we neglected monetary policy, which in the current economic context is not considered to be effective in the short run to face a shock of this amplitude, and we thus considered two policies: helicopter money for households and easy credit for firms. We find that some kind of easy credit is needed to avoid a wave of bankruptcies, and mixing both policies is effective, provided policy is strong enough. We also highlight that, for strong enough shocks, some flexibility on firm fragility might be needed for a long time (a few years) after the shock to prevent a second wave of bankruptcies [41] and removing support to firms too early before a robust recovery is underway can cause a rise in bankruptcies of illiquid but otherwise viable firms [42]. Too weak a policy intervention is not effective and can result in a "swoosh" recovery or no recovery at all. Our main message is that a threshold effect is at play, with potentially sharp changes in outcome upon small changes of policy strength.

As our model vividly illustrates, a key factor in permitting future recovery is to prevent any permanent losses in productive capacity. It is already feared that some firms and sectors facing difficulties may never recover, thus preventing the economy from going back to 100% capacity [29, 43]. Our results then suggest that governments should err on the side of caution and do "whatever it takes" to prevent the economy from falling into a bad state, supporting firms and households, and stimulate a rapid recovery. However, we find that when policy is successful, post-crisis inflation is significantly increased compared to the pre-crisis period. Nevertheless, this might be a reasonable cost to pay for faster recovery, especially because monetary policy (not considered here, but easy to implement [17, 18]) may in fact partially alleviate this effect.

Note that within the present work, a conscious decision has been taken to not model the dynamics of the pandemic along with the dynamics of the economy. The evolution of the pandemic is contingent not just on an optimal policy, but depends crucially on its timing, on society's compliance with that policy, on the availability of an efficient vaccine, etc. This work rather focuses on modeling the evolution of the economy itself, with the pandemic effectively modelled as one or several supply and demand shocks of different amplitudes and duration. Importantly, lifting of lockdowns in itself is not sufficient for the economy to recover since there is still widespread fear of the disease [29].

We want to again emphasize that the results and policy prescriptions presented here are only meant to be general and qualitative. Obviously, each country's policy response to the crisis should be adapted on a case by case basis. Nonetheless, we believe that our model captures the basic phenomenology of a Covid-19–like shock and is flexible enough to be used as an efficient "telescope for the mind" [21].

Many extensions are possible or in fact mandatory for Mark-0 to grasp further essential features of the real economy. For instance, introducing more explicitly a government or a public sector would be a useful extension of the current model, allowing for fiscal policy measures on top of emergency measures and monetary policy. As discussed above, we have not investigated in this study interest rate cuts or hikes. Whereas such cuts are not expected to play a major role in the short term management of the crisis, the effect of monetary policy on the long-term fate of the economy (in particular on inflation) can be easily examined within Mark-0, following Ref. [18].

Furthermore, one should note that in Mark-0 there is no splitting of debt into a public and a private sector, hence no competition between investments in corporate bonds and in government bonds. As a result, within the current framework we are unable to model panic effects that would result from substantial increases in public debt, which could possibly result in runaway public debt, confidence collapse and hyperinflation.

Finally, the level of heterogeneity in Mark-0 is somehow limited, because the input-output network of firms, and the power-law distribution of firm sizes, are not modeled, and a single representative household is present. The most needed extension for the current model is the introduction of inequalities—of firm sizes and of household wealth and wages. One of the peculiarities of the Covid-19 shock has been the asymmetry in the way the crisis has affected households, with lower spending by high-income households compounding the situation for low-income households [44]. A disparity in the outcomes following economic shock is already being seen with the prospect of a "K-shaped" recovery, with one section of the citizenry recovering quickly with the other struggling to recover [45]. Geographical heterogeneities within a country are also worth considering. As the pandemic has shown, certain regions can be more or less susceptible to economic and health risks [9], with consequences for the policy mix to be used. Taking into account such heterogeneities (income, region, economic sector, etc.), as in e.g. [24, 28], will indeed be an important step towards making Mark-0 a better representation of the real-world.

## Author Contributions

**Conceptualization:** Dhruv Sharma, Jean-Philippe Bouchaud, Stanislao Gualdi, Marco Tarzia, Francesco Zamponi.

**Data curation:** Dhruv Sharma.

**Formal analysis:** Dhruv Sharma.

**Methodology:** Jean-Philippe Bouchaud, Stanislao Gualdi, Marco Tarzia, Francesco Zamponi.

**Software:** Dhruv Sharma, Stanislao Gualdi, Francesco Zamponi.

**Visualization:** Dhruv Sharma, Stanislao Gualdi.

**Writing – original draft:** Dhruv Sharma, Jean-Philippe Bouchaud, Stanislao Gualdi, Marco Tarzia, Francesco Zamponi.

**Writing – review & editing:** Dhruv Sharma, Jean-Philippe Bouchaud, Stanislao Gualdi, Marco Tarzia, Francesco Zamponi.

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
