## [Decision Letter · Decision Letter 0]

14 Jan 2021

PONE-D-20-37904

V -, U -, L - or W-shaped recovery after COVID-19: Insights from an Agent Based Model

PLOS ONE

Dear Dr. Sharma,

Thank you for submitting your manuscript to PLOS ONE. After careful consideration, we feel that it has merit but does not fully meet PLOS ONE’s publication criteria as it currently stands. Therefore, we invite you to submit a revised version of the manuscript that addresses the points raised during the review process.

The manuscript requires further revisions regarding empirical research design, design, and policy implications.

We look forward to receiving your revised manuscript.

Kind regards,

Stefan Cristian Gherghina, PhD. Habil.

Academic Editor

PLOS ONE

Journal Requirements:

Reviewers' comments:

Reviewer's Responses to Questions

**Comments to the Author**

1. Is the manuscript technically sound, and do the data support the conclusions?

Reviewer #1: Yes

Reviewer #2: Partly

Reviewer #3: Yes

2. Has the statistical analysis been performed appropriately and rigorously? 

Reviewer #1: Yes

Reviewer #2: N/A

Reviewer #3: Yes

3. Have the authors made all data underlying the findings in their manuscript fully available?

Reviewer #1: Yes

Reviewer #2: No

Reviewer #3: Yes

4. Is the manuscript presented in an intelligible fashion and written in standard English?

Reviewer #1: Yes

Reviewer #2: Yes

Reviewer #3: Yes

5. Review Comments to the Author

Reviewer #1: The Results section should be presented at the end, after describing the model and all approximations. In this section, the authors describe the parameters of the model before describing the model itself.

99-106 and table 1 shows the empirical parameters of the model. However, their significance is not substantiated.

The Politics section offers different scenarios for overcoming the crisis, but without prior formalization of the mathematical model, it is impossible to understand these scenarios.

ABM models - depend on random values. Therefore, the simulation results may be unstable. It is not clear from Figures 4 and 5 this is the result of the work of an ensemble of models or a single simulation.

The authors do not provide an analysis of the sensitivity and stability of the model from changes in various factors.

The Model section should be moved to the beginning of the article.

In this section it is expedient to formalize the model in the form of the scheme of interaction of various components-blocks of system

It would be expedient to justify the choice of values of empirical parameters of the model to the characteristics of a particular country.

It would be interesting to compare the effectiveness of the strategies of different countries in comparison with the results of the model

Reviewer #2: The short title is better than long title that creates confusion.

The study is interesting with the aim to suggest guidelines for economic recovery after COVID-19 pandemic crisis and shock of lockdowns.

Introduction has to provide more theoretical background about COVID-19 and risk factors current and future in society to know the problem for health and economies (see suggested readings that have to be read and used in the text).

The study is based on simulations and without empirical evidence. Of course, the recovery depends on countries. I suggest to clarify better research setting because countries are different …a recovery in Italy is different from a recovery in Sweden because of different structural indicators…. for instance, Italy has a high public debt and a similar shock generates different effect than other countries. Moreover, also the lockdown generates different effects according to the duration…longer lockdown generates a higher deterioration of economic systems than short one (see new literature).

How these aspects are considered in the model?

The contraction of some economies is higher than others.

In short, there are manifold variables to control and I suggest whenever possible to discriminate the design of the model for countries that have some similarities, such as Spain and Italy, Germany and France, Scandinavian countries etc. to be more consistent with the reality and reduce the heterogeenity between countries and provide more reliable policy implications.

As you know the applications of policies generates different effects according to the countries and the structure of their economies… I would like to know how it is possible a match between these different shapes L, W, etc. of recovery and different countries, and policy implications…. I know it is necessary a time consuming revision, but authors can provide more indications to provide a main contribution for political economy of growth post COVID-19.

Conclusion has to present better manifold limitations of this study for the complexity of factors and heterogeneity of countries and in particular for the methods of inquiry based on simulations done in a computer lab.

Suggested readings of relevant papers that have to be read and all inserted in the text and references to improve the study.

Leach, M., MacGregor, H., Scoones, I., Wilkinson, A. 2021Post-pandemic transformations: How and why COVID-19 requires us to rethink development, World Development 138,105233

Coccia M., 2020. National lockdown to cope with COVID-19 pandemic: effects (contradictory) on public health and (negative) on economic system, Working Paper CocciaLab n. 56D/2020, CNR -- National Research Council of Italy. Available at Research Square, DOI is: 10.21203/rs.3.rs-115665/v1

Abuselidze, G., Slobodianyk, A. 2021 Pandeconomic crisis and its impact on small open economies: A case study of COVID-19 Advances in Intelligent Systems and Computing, 1258 AISC, pp. 718-728

Coccia M. 2020. An index to quantify environmental risk of exposure to future epidemics of the COVID-19 and similar viral agents: Theory and Practice. Environmental Research, Article number 110155, DOI: 10.1016/j.envres.2020.110155

Yoshino, N., Hendriyetty, N. 2020 The COVID-19 Crisis: Policy Recommendations for Japan. Economists' Voice , 17(1),20200017

Iuga, I.C., Mihalciuc, A. 2020Major crises of the XXIst century and impact on economic growth. Sustainability (Switzerland), 12(22),9373, pp. 1-20

Reviewer #3: The article is well written and provides interesting insights about the response of the economy to the pandemic due to COVID-19. In general, the authors did a good job explaining the purpose of the study, and described properly the instruments they used to develop the study. I have some minor comments for the authors to be addressed before the study can be accepted for publication.

1. I suggest including some literature in the introduction (Lines 40-44) about using ABM to understand the impacts of COVID. For example, please review the following article.

https://doi.org/10.1016/j.ssci.2020.105022

2. Line 204

The authors could explain better why they selected 2 policies. Are there different policies that simulate the same effects from the policies that the authors used in the study in the literature? If so, explain why other policies are not selected.

3. I suggest to the authors to include in the study some discussion about how the results were validated and verified. If not, please also state that clearly.

4. I would also recommend to the authors to include the word economy in the title.

6. PLOS authors have the option to publish the peer review history of their article (what does this mean?). If published, this will include your full peer review and any attached files.

Reviewer #1: **Yes: **Prof. Yaroslav Vyklyuk

Reviewer #2: No

Reviewer #3: No

---

## [Author Response · Author response to Decision Letter 0]

4 Feb 2021

We thank the reviewers for their comments which have helped to improve the manuscript. All points raised by the reviewers have been addressed in the Rebuttal letter attached with the resubmission.

---

## [Decision Letter · Decision Letter 1]

15 Feb 2021

V -, U -, L - or W-shaped economic recovery after COVID-19: Insights from an Agent Based Model

PONE-D-20-37904R1

Dear Dr. Sharma,

We’re pleased to inform you that your manuscript has been judged scientifically suitable for publication and will be formally accepted for publication once it meets all outstanding technical requirements. In this vein, there are required a couple of revisions regarding references.

Kind regards,

Stefan Cristian Gherghina, PhD. Habil.

Academic Editor

PLOS ONE

Additional Editor Comments (optional):

Reviewers' comments:

Reviewer's Responses to Questions

**Comments to the Author**

1. If the authors have adequately addressed your comments raised in a previous round of review and you feel that this manuscript is now acceptable for publication, you may indicate that here to bypass the “Comments to the Author” section, enter your conflict of interest statement in the “Confidential to Editor” section, and submit your "Accept" recommendation.

Reviewer #1: All comments have been addressed

Reviewer #2: All comments have been addressed

Reviewer #3: All comments have been addressed

2. Is the manuscript technically sound, and do the data support the conclusions?

Reviewer #1: Yes

Reviewer #2: Yes

Reviewer #3: Yes

3. Has the statistical analysis been performed appropriately and rigorously? 

Reviewer #1: Yes

Reviewer #2: Yes

Reviewer #3: Yes

4. Have the authors made all data underlying the findings in their manuscript fully available?

Reviewer #1: Yes

Reviewer #2: Yes

Reviewer #3: Yes

5. Is the manuscript presented in an intelligible fashion and written in standard English?

Reviewer #1: Yes

Reviewer #2: Yes

Reviewer #3: Yes

6. Review Comments to the Author

Reviewer #1: The author took into account the comments of the reviewer. Therefore, I believe that the article can be published in a journal

Reviewer #2: V -, U -, L - or W-shaped economic recovery after COVID-19: Insights from an Agent Based Model

I have read thoroughly the revised version of paper. The authors have done considerable additional work, and addressed all concerns and criticisms in the revised manuscript, which I believe has improved substantially in the theoretical framework, study design and discussion of results.

Finally, I suggest to revise some citations of papers to update them and correctly cite some articles in references, such as:

--Fornaro, Luca and Wolf, Martin, 2020. Covid-19 Coronavirus and Macroeconomic Policy (March 2020). CEPR Discussion Paper No. DP14529, Available at SSRN: https://ssrn.com/abstract=3560337

--Coccia M. 2021. The relation between length of lockdown, numbers of infected people and deaths of Covid-19, and economic growth of countries: Lessons learned to cope with future pandemics similar to Covid-19. Science of The Total Environment, Available online 12 February 2021, 145801. https://doi.org/10.1016/j.scitotenv.2021.145801

--Eichenbaum M. S., Rebelo S., Trabandt M., 2020. The Macroeconomics of Epidemics, NBER Working Papers 26882, National Bureau of Economic Research, Inc. doi = 10.3386/w26882

After that, the paper is OK.

Reviewer #3: Thank you for preparing a response to my comments. Since this is a very current and interesting topic, I recommended the article for publication.

7. PLOS authors have the option to publish the peer review history of their article (what does this mean?). If published, this will include your full peer review and any attached files.

Reviewer #1: No

Reviewer #2: No

Reviewer #3: No

---

## [Editor Report · Acceptance letter]

19 Feb 2021

PONE-D-20-37904R1 

V–, U–, L– or W–shaped economic recovery after Covid-19: Insights from an Agent Based Model 

Dear Dr. Sharma:

I'm pleased to inform you that your manuscript has been deemed suitable for publication in PLOS ONE. Congratulations! Your manuscript is now with our production department. 

Kind regards, 

on behalf of

Dr. Stefan Cristian Gherghina 

Academic Editor

PLOS ONE